# Sequence signatures of two public antibody clonotypes that bind SARS-CoV-2 receptor binding domain

Timothy J. C. Tan [1,9], Meng Yuan [2,9], Kaylee Kuzelka[3], Gilberto C. Padron[3], Jacob R. Beal[3], Xin Chen[1], Yiquan Wang [3], Joel Rivera-Cardona[4], Xueyong Zhu [2], Beth M. Stadtmueller[3], Christopher B. Brooke[4,5], Ian A. Wilson [2,6,7,8,10 ✉] & Nicholas C. Wu [1,3,5,10 ✉]

Since the COVID-19 pandemic onset, the antibody response to SARS-CoV-2 has been extensively characterized. Antibodies to the receptor binding domain (RBD) on the spike protein are frequently encoded by IGHV3-53/3-66 with a short complementarity-determining region (CDR) H3. Germline-encoded sequence motifs in heavy chain CDRs H1 and H2 have a major function, but whether any common motifs are present in CDR H3, which is often critical for binding specificity, is not clear. Here, we identify two public clonotypes of IGHV3-53/3-66 RBD antibodies with a 9-residue CDR H3 that pair with different light chains. Distinct sequence motifs on CDR H3 are present in the two public clonotypes that seem to be related to differential light chain pairing. Additionally, we show that Y58F is a common somatic hypermutation that results in increased binding affinity of IGHV3-53/3-66 RBD antibodies with a short CDR H3. These results advance understanding of the antibody response to SARS-CoV-2.

[1] Center for Biophysics and Quantitative Biology, University of Illinois at Urbana-Champaign, Urbana, IL, USA. [2] Department of Integrative Structural and Computational Biology, The Scripps Research Institute, La Jolla, CA, USA. [3] Department of Biochemistry, University of Illinois at Urbana-Champaign, Urbana, IL, USA. [4] Department of Microbiology, University of Illinois at Urbana-Champaign, Urbana, IL, USA. [5] Carl R. Woese Institute for Genomic Biology, University of Illinois at Urbana-Champaign, Urbana, IL, USA. [6] The Skaggs Institute for Chemical Biology, The Scripps Research Institute, La Jolla, CA, USA. [7] IAVI Neutralizing Antibody Center, The Scripps Research Institute, La Jolla, CA, USA. [8] Consortium for HIV/AIDS Vaccine Development (CHAVD), The Scripps Research Institute, La Jolla, CA, USA. [9] These authors contributed equally: Timothy J.C. Tan, Meng Yuan. [10] These authors jointly supervised this work: Ian A. Wilson, Nicholas C. Wu. ✉email: wilson@scripps.edu; nicwu@illinois.edu

Severe acute respiratory syndrome coronavirus-2 (SARS-CoV-2) is the etiological agent of coronavirus disease 2019 (COVID-19)[1,2], which primarily results in respiratory distress, cardiac failure, and renal injury in the most severe cases[3,4]. The virion is decorated with the spike (S) glycoprotein, which contains a receptor-binding domain (RBD) that mediates virus entry by binding to angiotensin-converting enzyme-2 (ACE-2) receptor on the surface of host cells[1,5–7]. To mitigate the devastating social and economic consequences of the pandemic, vaccines and post-exposure prophylaxes including antibody cocktails that exploit reactivity to the S protein are being developed at an unprecedented rate. Several vaccines are currently in various stages of clinical trials[8,9]. Most notable are the mRNA vaccines from Pfizer-BioNTech and Moderna, which have been issued emergency use authorization by the Food and Drug Administration for distribution in the United States[10–12], the adenovirus-vectored DNA vaccine from Johnson & Johnson[13,14], and the Oxford-AstraZeneca chimpanzee adenovirus-vectored DNA vaccine in the United Kingdom[15–17]. In humans, most neutralizing antibodies to SARS-CoV-2 target the immunodominant RBD on the S protein[18,19], and can abrogate virus attachment and entry into host cells[20,21]. In the past year, many RBD antibodies have been isolated and characterized from convalescent SARS-CoV-2 patients[22–42].

Antibody diversity is generated through V(D)J recombination[43–45]. Three genes, one from each of the variable (V), diversity (D), and joining (J) loci, are combined to form the coding region for the heavy chain. In humans, genes encoding for the V, D, and J regions are denoted as *IGHV*, *IGHD* and *IGHJ*, respectively. Two complementarity-determining regions on the heavy chain (CDRs H1 and H2) are encoded by the V gene while the third (CDR H3) is encoded by the V(D)J junction. A similar process occurs in assembly of the coding region for the light chain except that the D gene is absent. The light chain genes also encode kappa and lambda chains that are denoted as *IGKV* and *IGKJ*, as well as *IGLV* and *IGLJ*, respectively. To further improve the affinity of antibodies to an antigen, affinity maturation occurs in vivo via somatic hypermutation (SHM)[46,47]. V(D)J recombination and SHM, therefore, ensure a diverse repertoire of antibodies is available for an immune response to the enormous number and variety of potential antigens.

Notwithstanding this antibody diversity, some RBD antibodies with strikingly similar sequences have been found in multiple convalescent SARS-CoV-2 patients[34,48,49]. These antibodies can be classified as public clonotypes if they share the same IGHV gene with similar CDR H3 sequences[50–54]. Over the past decade, public clonotypes to human immunodeficiency virus[50], malaria[54], influenza[51], and dengue virus[55] have been discovered. Antibodies to SARS-CoV-2 RBD frequently use IGHV3-53 and IGHV3-66[25,33,49,56], which only differ by one amino acid (i.e. I12 in IGHV3-53 and V12 in IGHV3-66). IGHV3-53/3-66 antibodies carry germline-encoded features that are critical for RBD binding—an NY motif in CDR H1 and an SGGS motif in CDR H2[33,49,56]. Nevertheless, IGHV3-53/3-66 RBD antibodies have varying lengths of CDR H3 with diverse sequences, which seem to deviate from the canonical definition of a public clonotype.

By categorizing IGHV3-53/3-66 RBD antibodies based on CDR H3 length and light chain usage, we now report two public clonotypes of IGHV3-53/3-66 RBD antibodies, both of which have a CDR H3 length of 9 amino acids (Kabat numbering) but with distinct sequence motifs. Structural and biochemical analyses show that these sequence motifs on CDR H3 are associated with light chain pairing preference. We also identify Y58F as a signature SHM among IGHV3-53/3-66 RBD antibodies that have a CDR H3 length of less than 15 amino acids. As the COVID-19 pandemic continues, knowledge of public antibodies against SARS-CoV-2 can inform on therapeutic development as well as vaccine assessment.

## Results

**Two public clonotypes of IGHV3-53/3-66 RBD antibodies.** In this study, we define clonotypic IGHV3-53/3-66 RBD antibodies as antibodies that share the same *IGL(K)V* genes and with identical CDR H3 length. Literature mining of 214 published IGHV3-53/3-66 RBD antibodies obtained from convalescent patients (Supplementary Data 1) revealed that the two most common clonotypes have a CDR H3 length of 9 amino acids and are paired with light chains IGKV1-9 (clonotype 1) and IGKV3-20 (clonotype 2), respectively (Fig. 1a). Antibodies from clonotype 1 have been observed across 10 studies[24–26,34–38,42], whereas antibodies from clonotype 2 are found across seven studies[24,26,34–36,39,42]. Interestingly, sequence logos revealed distinct sequence features of CDR H3 between clonotype 1 and clonotype 2 antibodies (Fig. 1b).

We further investigated *IGHJ* gene usage in the two major clonotypes of IGHV3-53/3-66 RBD antibodies. Among the IGHV3-53/3-66 RBD antibodies with a CDR H3 length of 9 amino acids, we observed a statistically significant bias in *IGHJ* gene usage ($p$-value = 2e-6, Fisher's exact test), where clonotypes 1 and 2 preferentially pair with IGHJ6 and IGHJ4, respectively (Fig. 1c). In fact, IGHJ6 encodes the last four amino acids (GMDV) in CDR H3 that are highly conserved in clonotype 1 (Fig. 1d, Supplementary Fig. 1a). Similarly, IGHJ4 encodes the last four amino acids (YFDY) in CDR H3 that are highly conserved in clonotype 2 (Fig. 1d, Supplementary Fig. 1b). Taken together, we demonstrate that IGHV3-53/3-66 RBD antibodies can be categorized into at least two public clonotypes.

**Structural analysis of signature motifs on CDR H3.** We further investigated sequence signatures of CDR H3s in clonotypes 1 and 2 (Fig. 1b). In particular, we focused on amino acid residues 96, 98, and 100 in CDR H3 since these residues show clear patterns of differential amino-acid preference between clonotype 1 and clonotype 2 antibodies. Subsequently, analysis was performed on structures of BD-604 (PDB 7CH4) and CC12.1 (PDB 6XC2), which are two clonotype 1 antibodies, as well as BD-629 (PDB 7CH5) and CC12.3 (PDB 6XC4), which are two clonotype 2 antibodies.

Residue 96 is usually Leu in clonotype 1 antibodies, whereas an aromatic residue, usually Tyr, occupies residue 96 in clonotype 2 antibodies. While $V_H$ L96 interacts with Y489 of the RBD in clonotype 1 antibodies via van der Waals interactions, $V_H$ F/Y96 is located at the center of a π−π stacking network that involves F456, Y489 and $V_H$ Y100 (Fig. 2a, b, Supplementary Fig. 2a, b; left panels). Substituting $V_H$ L96 in clonotype 1 with Y96 would result in a clash with RBD Y489, whereas substituting $V_H$ F/Y96 in clonotype 2 with L96 would abolish the π−π stacking network but still maintain a hydrophobic core.

Residue 98 in CDR H3 of clonotype 1 antibodies does not show a strong amino-acid preference, since it is located in a relatively open space in the RBD interface (Figs. 1b, 2a, Supplementary Fig. 2a; middle panels). On the other hand, a highly conserved acidic residue at position 98 in the CDR H3 loop of clonotype 2 antibodies contributes to formation of hydrogen bond interactions with $V_H$ Y52 as well as electrostatic interactions with RBD K417 and $V_L$ R96 (Fig. 2b, Supplementary Fig. 2b; middle panels). Consistently, $V_L$ R96 is highly conserved in clonotype 2 antibodies, but not in other IGHV3-53/3-66 RBD antibodies (Supplementary Fig. 3). Thus, the electrostatic interactions between $V_H$ D/E98 and $V_L$ R96 are highly conserved in clonotype 2 antibodies and can likely help stabilize the CDR H3 loop

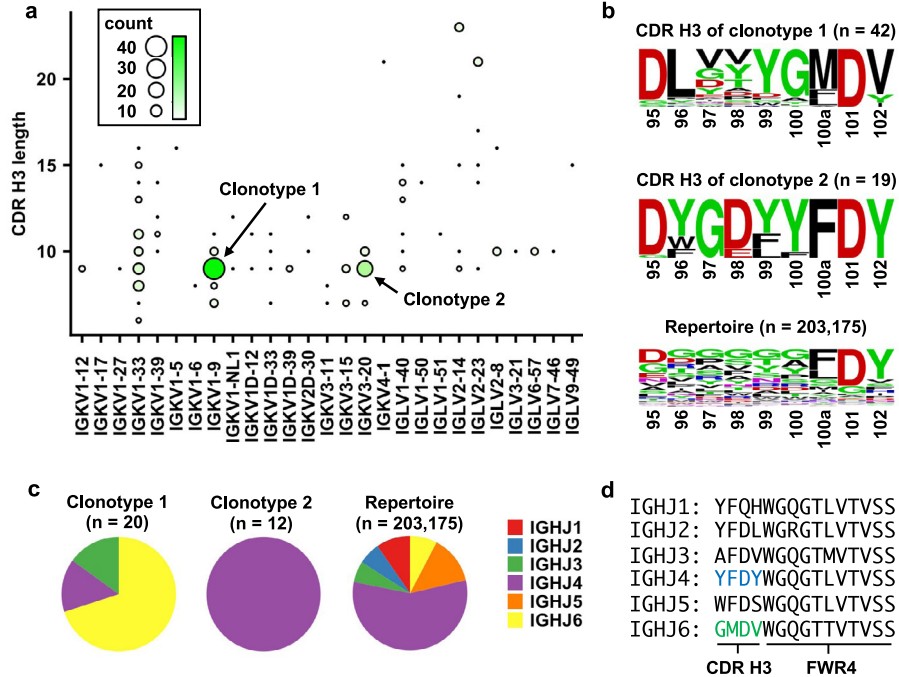

**Fig. 1 Two major clonotypes of IGHV3-53/3-66 antibodies to SARS-CoV-2 RBD. a** The number of IGHV3-53/3-66 RBD antibodies that use the same light chain with the same CDR H3 are tabulated. The two most common combinations are IGKV1-9 pairing with 9 aa CDR H3 and IGKV3-20 pairing with 9 aa CDR H3, denoted as clonotype 1 and clonotype 2, respectively. **b** Sequence logos for the CDR H3 regions of IGHV3-53/3-66 antibodies that pair with IGKV1-9 or IGKV3-20. A sequence logo for the CDR H3 regions of 203,175 IGHV3-53/3-66 antibodies from Observed Antibody Space database[91] that have a CDR H3 length of 9 aa is shown for reference (repertoire). The position of each residue is labeled on the *x*-axis based on Kabat numbering. **c** *IGHJ* gene usage for clonotypes 1 and 2, as well as 203,175 IGHV3-53/3-66 antibodies from Observed Antibody Space database that have a CDR H3 length of 9 aa (repertoire), are shown as pie charts. For antibodies in clonotypes 1 and 2, only those with nucleotide sequence information available were analyzed. **d** Amino acid sequences for different *IGHJ* genes are shown. Source data are available in the Source data file.

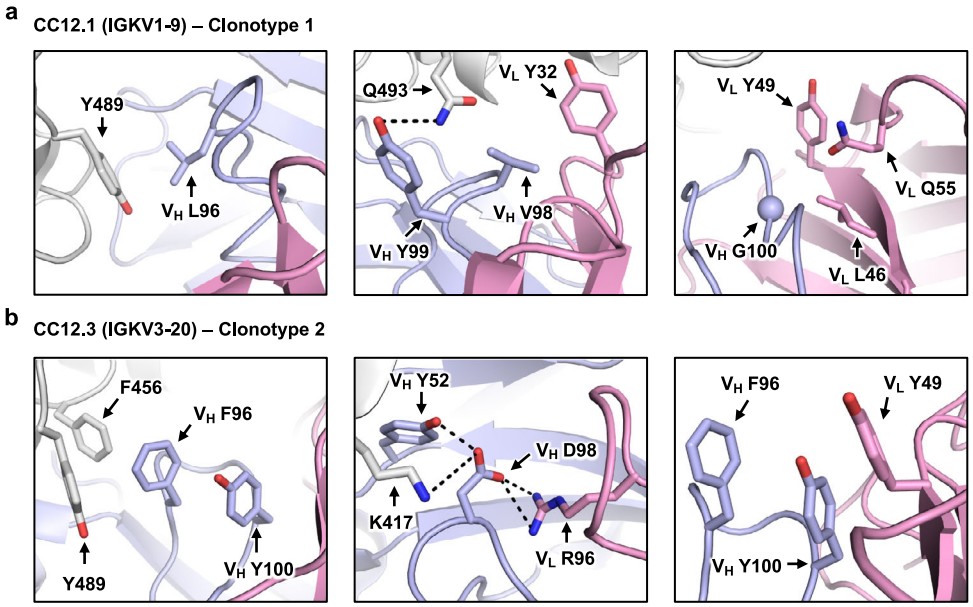

**Fig. 2 Structural analysis of sequence signatures in CDR H3 of clonotypes 1 and 2. a** Interaction of L96, V98, and G100 (Kabat numbering) in CDR H3 of CC12.1 (PDB 6XC2) with the IGKV1-9 light chain of the antibody, and SARS-CoV-2 RBD. **b** Interaction of F96, D98 and Y100 (Kabat numbering) in CDR H3 of CC12.3 (PDB 6XC4) with the IGKV3-20 light chain of the antibody, and SARS-CoV-2 RBD. Gray: RBD; light blue: heavy chain; pink: light chain. V_H and V_L indicate residues belong to the heavy and light chain of the antibody, respectively.

conformation to minimize entropic cost upon binding to SARS-CoV-2 RBD.

Residue 100 is usually Gly in CDR H3 of clonotype 1 antibodies (Fig. 1b). Structural analysis shows that small, non-polar amino acids are favored at position 100 due to the limited space around that residue (Fig. 2a, Supplementary Fig. 2a; right panels). Moreover, G100 in clonotype 1 has a positive Φ angle, which is typically less favorable for non-Gly amino acids. In contrast,

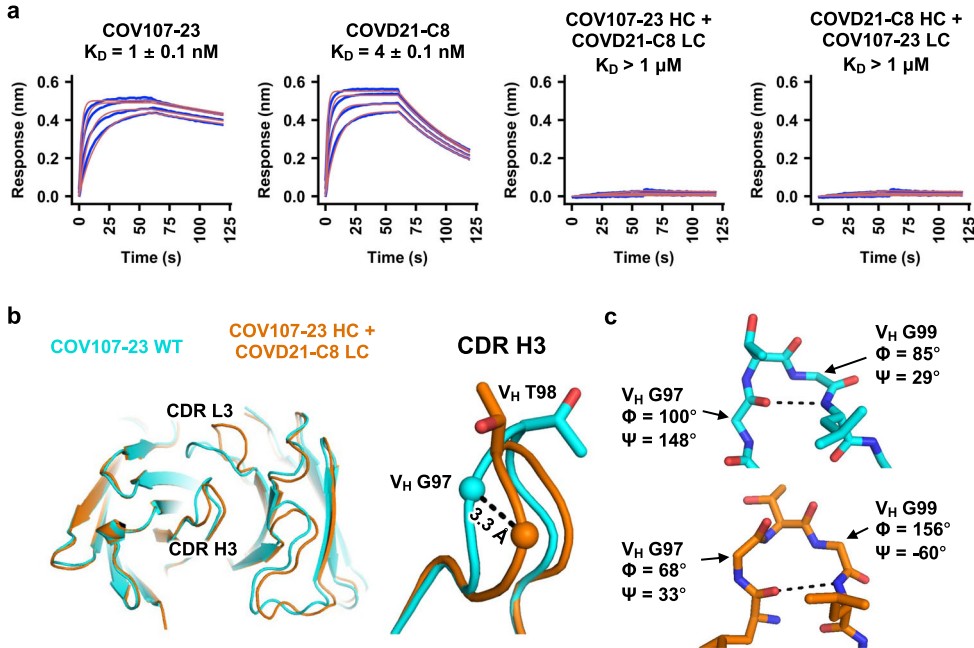

**Fig. 3 Specific pairing of CDR H3 and light chain is critical for IGHV3-53/3-66 antibody binding to SARS-CoV-2 RBD. a** Binding of different Fabs to SARS-CoV-2 RBD was measured by biolayer interferometry with RBD loaded onto the biosensor and Fab in solution. Y-axis represents the response. Dissociation constant ($K_D$) for each Fab was obtained using a 1:1 binding model, which is represented by the red curves. COV107-23 belongs to clonotype 1, whereas COVD21-C8 belongs to clonotype 2. **b** Fab crystal structures of wild-type (WT) COV107-23 and COV107-23 heavy chain pairing with COVD21-C8 light chain are compared. Left panel: structural alignment using residues 1–90 of the heavy chain. Right panel: Zoomed-in view for the CDR H3. **c** Conformations at the tips of the CDR H3s in WT COV107-23 and COV107-23 heavy chain pairing with COVD21-C8 light chain are shown. A β-turn is observed in CDR H3 of WT COV107-23, with $V_H$ G97 and $V_H$ G99 at the i and i + 2 positions, respectively. $V_H$ indicates residues belong to the heavy chain.

residue 100 is a highly conserved Tyr in CDR H3 of clonotype 2 antibodies (Fig. 1b). Structural analysis shows that $V_H$ Y100 contributes to the π−π stacking network that is formed via the aromatic ring at $V_H$ residue 96 (see above) and an aromatic residue at $V_L$ residue 49 (Fig. 2b, Supplementary Fig. 2b; right panels).

Additionally, we investigated the structural basis of the conservation of $V_H$ Y102 among clonotype 2 antibodies. Structural analysis reveals that $V_H$ Y102 interacts with RBD Y486 via π−π interactions (Supplementary Fig. 4). Only IGHJ4 offers a bulky aromatic side chain at residue 102 (Fig. 1d), which explains the common usage of IGHJ4 in clonotype 2 antibodies. In contrast, clonotype 1 antibodies frequently use IGHJ6 (Fig. 1d), which has a much shorter Val at residue 102, most likely because IGHJ6 encodes a Gly at residue 100 that can avoid steric clashes with the light chain (see above, Fig. 2a, Supplementary Fig. 2a; right panels). Of note, the only other *IGHJ* gene that encodes a non-bulky amino acid at residue 100 is IGHJ3 (Ala). IGHJ1, IGHJ2, IGHJ4, and IGHJ5 all encode a bulky residue at residue 100 (Fig. 1d), which may be disfavored in clonotype 1 antibodies due to the limited space where $V_H$ residue 100 is located, as demonstrated in our simulations using Rosetta (Supplementary Fig. 5). Overall, our structural analyses provide a structural basis for the differential signature sequence motifs in CDR H3 between clonotype 1 and clonotype 2 antibodies.

## Incompatibility of CDR H3 between clonotype 1 and clonotype 2 antibodies.
To understand the influence of light-chain usage in CDR H3 sequences, we performed a structural alignment of RBD-bound CDR H3 from two clonotype 1 antibodies, namely BD-604 and CC12.1, and two clonotype 2 antibodies, namely BD-629 and CC12.3 (Supplementary Fig. 2c–f). While the CDR H3

conformations are similar within each clonotype (Cα RMSD ranges from 0.27 to 0.41 Å), they are quite different between clonotypes (Cα RMSD ranges from 0.77 Å to 1.5 Å). Although our sample size is small, this analysis suggests that antibodies from clonotypes 1 and 2 have different preferences for their CDR H3 conformations. This differential preference may be partly influenced by light-chain usage, as indicated by the structural analyses above on $V_H$ residues 96, 98, and 100 (Fig. 2, Supplementary Figs. 2 and 5).

To experimentally examine the compatibility between CDR H3 and the light chains from clonotype 1 and clonotype 2 antibodies, we focused on antibodies COV107-23 (clonotype 1) and COVD21-C8 (clonotype 2). The heavy-chain sequences of these two antibodies only differ by four amino acids in CDR H3, namely $V_H$ residues 96, 98, 99, and 100 (Supplementary Fig. 6a). Of note, COV107-23 uses IGHJ4, which is seldom observed among clonotype 1 antibodies but highly preferred in clonotype 2 antibodies (Fig. 1c), to encode the two amino acids at the C-terminus of its CDR H3 (Supplementary Fig. 6b). Both COV107-23 and COVD21-C8 bind strongly to the SARS-CoV-2 RBD, with dissociation constants ($K_D$) of 1 nM and 4 nM, respectively (Fig. 3a). However, when their light chains are swapped, their binding affinity to the RBD is weakened substantially to $K_D > 1$ μM. We further determined apo crystal structures of COV107-23 paired with its native light chain and with the light chain from COVD21-C8 to 2.0 Å and 3.3 Å, respectively (Supplementary Table 1). The conformations of CDR H3 indeed differ when paired with different light chains, as exemplified by the 3.3 Å displacement of $V_H$ G97 near the tip of CDR H3 and different side-chain orientations of $V_H$ T98 (Fig. 3b). In addition, a type I′ β-turn is observed at the tip of CDR H3 in COV107-23 when paired with its native light chain but not with the light chain from COVD21-C8 (Fig. 3c). These observations demonstrate that the

conformation of CDR H3 changes substantially when IGKV1-9 in COV107-23 is swapped to IGKV3-20, which abolishes the binding to RBD (Fig. 3a). The CDR H3 conformation is therefore a determinant for compatibility between the CDR H3 sequence and the light chain in IGHV3-53/3-66 RBD antibodies.

**Compatibility of different CDR H3 variants with IGKV1-9 for binding to RBD.** Besides antibodies from clonotypes 1 and 2, other IGHV3-53/3-66 RBD antibodies with a range of CDR H3 lengths pair with different light chains (Fig. 1a). We further aimed to expand our analysis on CDR H3 compatibility to include CDR H3 from IGHV3-53/3-66 RBD antibodies other than clonotypes 1 and 2. In particular, we focused on identifying CDR H3 sequences that are compatible with IGKV1-9, which is used by clonotype 1 antibodies for binding to RBD, because IGKV1-9 is the most commonly used light chain gene among IGHV3-53/3-66 RBD antibodies (Supplementary Data 1) and clonotype 1 is the most predominant clonotype (Fig. 1a). We first compiled a list of 143

CDR H3 variants that were observed in IGHV3-53/3-66 RBD antibodies (Supplementary Data 1). A yeast display library was then constructed with these 143 CDR H3 variants in the B38 antibody, which is a IGHV3-53/IGKV1-9 RBD antibody[28]. Subsequently, fluorescence-activated cell sorting (FACS) was performed on the yeast display library based on antibody expression level and binding to SARS-CoV-2 RBD (Supplementary Fig. 7). The enrichment level of each CDR H3 variant in the sorted library was quantified by next-generation sequencing (see Methods, Supplementary Fig. 8 and Supplementary Data 2). CDR H3 variants that were positively enriched in binding ($\log_{10}$ enrichment >0) are derived from both IGKV1-9 and non-IGKV1-9 antibodies (Fig. 4a). The native CDR H3 for B38 has a $\log_{10}$ enrichment level of -0.002. As a result, positively enriched CDR H3 variants should have a higher affinity than wild-type B38. A total of 68% (17 out of 25) binding-enriched CDR H3 variants have a length of 9 amino acids, whereas only 31% (37 out of 118) have a length of 9 amino acids in the non-enriched group (Fig. 4b). Interestingly, binding-enriched CDR H3 variants with a

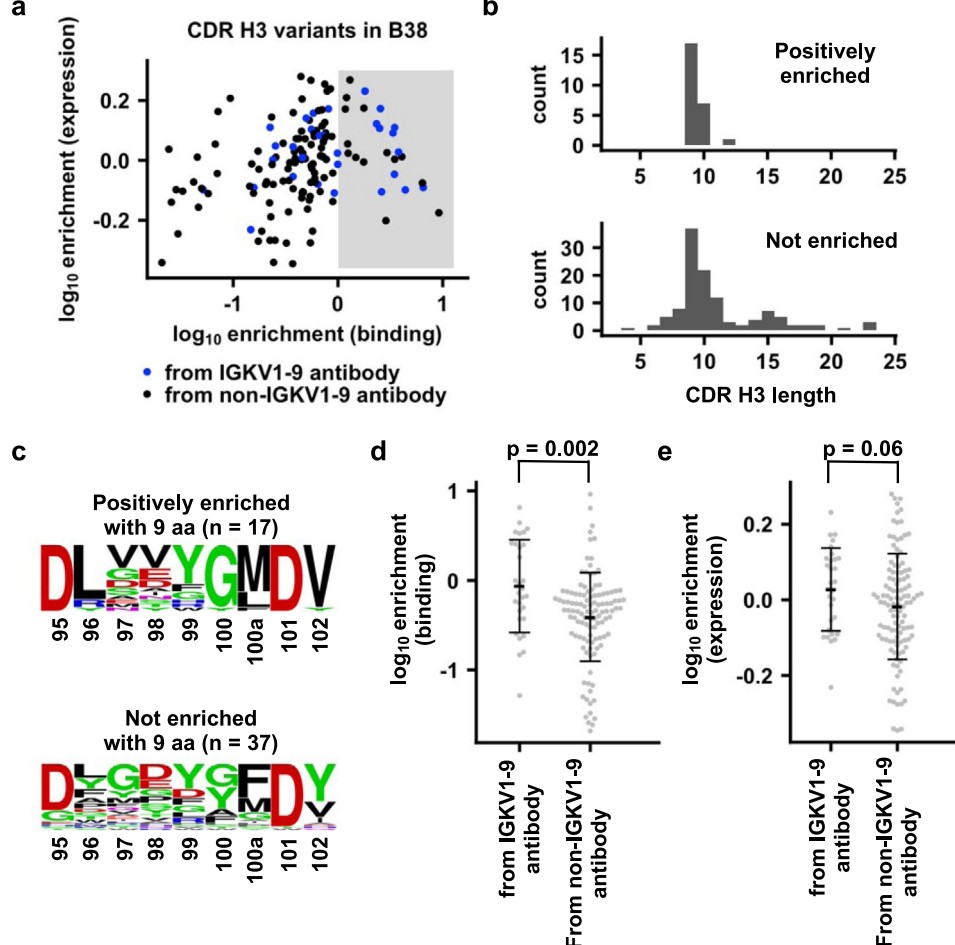

**Fig. 4 Binding and expression profiling of 143 CDR H3 variants in B38 antibody. a** For each of the 143 CDR H3 variants, the enrichment in occurrence frequencies after FACS selections for binding to RBD and expression level are shown. Blue: CDR H3 variants that are derived from IGHV3-53/3-66 RBD antibodies that use IGKV1-9. Black: CDR H3 variants that are derived from IGHV3-53/3-66 RBD antibodies that do not use IGKV1-9. Shaded area indicates $\log_{10}$ enrichment in binding >0. Data are from the average of $n = 2$ independent biological replicates. **b** The amino-acid length distribution of CDR H3 variants that are positively enriched in binding ($\log_{10}$ enrichment in binding >0) or not ($\log_{10}$ enrichment in binding ≤0) is shown. **c** Sequence logos are shown for CDR H3 variants with 9 aa (Kabat numbering) that are positively enriched or not enriched. **d** Comparison of $\log_{10}$ enrichment in binding for CDR H3 variants from IGHV3-53/3-66 RBD antibodies that use IGKV1-9 and those that do not use IGKV1-9. **e** Comparison of $\log_{10}$ enrichment in expression for CDR H3 variants from IGHV3-53/3-66 RBD antibodies that use IGKV1-9 and those that do not use IGKV1-9. **d**, **e** Two-tailed Student's $t$-test was used to compute the $p$-value. Error bars represent standard deviations. The center horizontal bars represent the means. Enrichment in expression and binding, as well as count data are provided in Supplementary Data 2. Data are from the average of $n = 2$ independent biological replicates. Source data are provided in the Source data file.

length of 9 amino acids displayed very similar sequence features as that of clonotype 1 antibodies obtained from literature mining (Figs. 1b and 4c). Of note, 41% (7 out of 17) binding-enriched CDR H3 variants with a length of 9 amino acids come from non-IGKV1-9 antibodies. Overall, our yeast display screen indicates that certain CDR H3s from non-IGKV1-9 RBD antibodies are compatible with IGKV1-9 for RBD binding and have similar sequence features as those CDR H3s from clonotype 1 antibodies.

We noticed that some CDR H3 sequences that come from IGKV1-9 RBD antibodies do not enrich in binding. One possibility is that they are still able to bind to RBD, but with a lower affinity than B38, which has a $K_D$ of 70 nM to the RBD[28]. However, as shown by our yeast display screen, CDR H3 sequences from IGKV1-9 antibodies in general have a significantly stronger binding to RBD than those from non-IGKV1-9 antibodies ($p$-value = 0.002, Fig. 4d), whereas their expression level is only marginally higher than that from non-IGKV1-9 antibodies ($p$-value = 0.06, Fig. 4d). Of note, our previous work has shown that binding affinity correlates well with neutralization activity for antibodies that bind to the epitope of B38 (i.e., epitope RBD-A, see Fig. 4G in Rogers et al.[26]). In addition, the binding affinity and neutralization activity of five clonotype 1 antibodies from Cao et al.[25] show a high correlation ($R = 0.86$, Supplementary Fig. 9). As a result, although the neutralization potency of B38 variants was not measured in this study, B38 variants with higher binding affinity would likely result in higher neutralization potency.

**Y58F is a signature SHM in IGHV3-53/3-66 RBD antibodies.** We further aimed to understand if there are common SHMs among IGHV3-53/3-66 RBD antibodies. We first categorized IGHV3-53/3-66 RBD antibodies from convalescent SARS-CoV-2 patients by CDR H3 length. The occurrence frequencies of individual SHMs in each category were then analyzed (Fig. 5a). This analysis included 165 IGHV3-53/3-66 RBD antibodies that

have sequence information available. One clear observation is that Y58F is highly common among IGHV3-53/3-66 RBD antibodies with a CDR H3 length of less than 15 amino acids, but completely absent when the CDR H3 length is 15 amino acids or above, suggesting that Y58F improves the binding affinity of IGHV3-53/3-66 antibodies to RBD only when they have a short CDR H3 loop (CDR H3 < 15 amino acids). To understand the effect of Y58F on binding of IGHV3-53/3-66 antibodies to the RBD, we compared the binding affinity of the same antibodies that carry either Y58 or F58 to the RBD. In particular, we focused on three IGHV3-53/3-66 RBD antibodies that have a CDR H3 length of 9 amino acids—one in clonotype 1 (COV107-23), and two in clonotype 2 (COVD21-C8 and CC12.3). Our biolayer interferometry (BLI) experiments showed that the Y58F mutation dramatically improved the affinity of the three antibodies (COV107-23, COVD21-C8, and CC12.3) by ~10-fold to ~1000-fold (Figs. 3a, 5b, Supplementary Fig. 10). As a control, we also performed the same experiment on an IGHV3-53/3-66 antibody with a CDR H3 length of 15 amino acids, namely COVA2-20. In contrast to those three IGHV3-53/3-66 RBD antibodies with a short CDR H3, COVA2-20 shows similar binding affinity to RBD between Y58 and F58 variants (Fig. 5b, Supplementary Fig. 10). Taken together, our results show that Y58F appears to be a signature SHM in IGHV3-53/3-66 RBD antibodies with CDR H3 length of < 15 amino acids. In fact, the results here are consistent with our previous finding that IGHV3-53/3-66 RBD antibodies with CDR H3 length of 15 amino acids or longer generally adopt a different binding mode as compared to those with a shorter CDR H3[56].

Interestingly, a Y58F mutation results in a loss of hydrogen bonding interactions between residue 58 of the heavy chain and T415 of the RBD (Supplementary Fig. 11), yet the mutation significantly increases the binding affinity of the antibody to the RBD. We then performed a structural analysis on seven IGHV3-53/3-66 RBD antibodies with a Y58F mutation and nine without[28,31,40,42,49,56–59]. Our results indicate that, upon removal

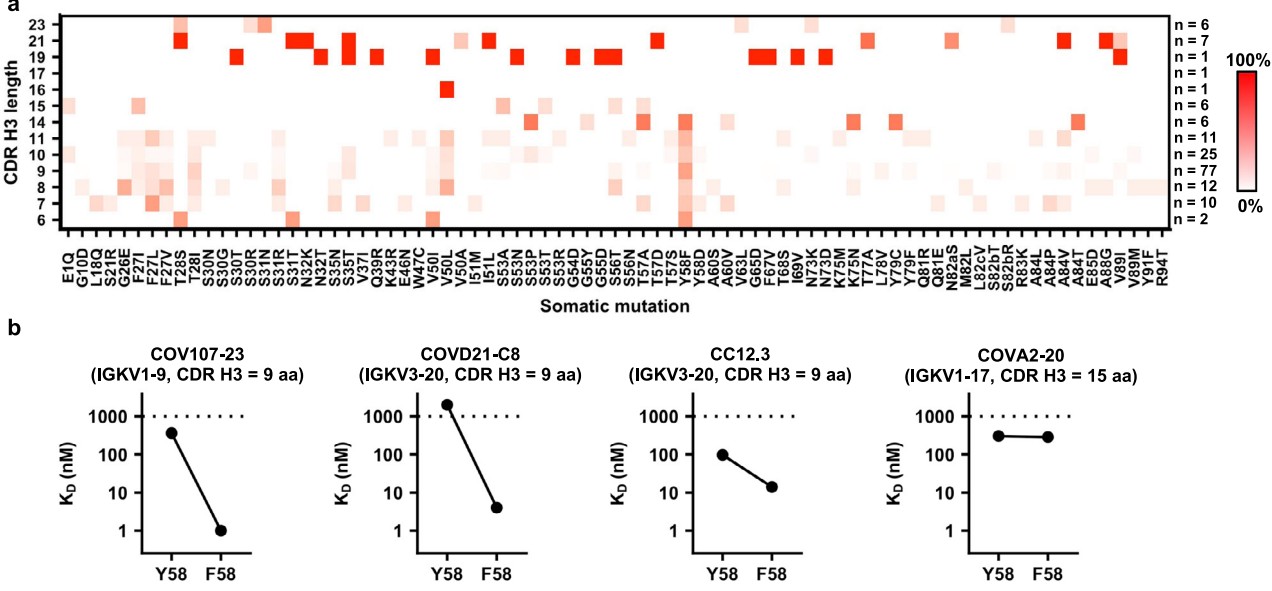

**Fig. 5 Y58F is a signature somatic hypermutation in IGHV3-53/3-66 RBD antibodies with a short CDR H3. a** IGHV3-53/3-66 RBD antibodies are categorized based on their CDR H3 length (Kabat numbering). Occurrence frequencies of individual somatic hypermutations in different categories were quantified and shown as a heatmap. The number of antibodies in each category is indicated on the right of the heatmap. **b** Both Y58 and F58 variants were constructed for four IGHV3-53 antibodies. Binding kinetics of each of these antibodies as Fab format to SARS-CoV-2 RBD was measured by biolayer interferometry with RBD loaded on the biosensor and Fab in solution. Y-axis represents dissociation constants ($K_D$) that were obtained using a 1:1 binding model. Of note, the WTs of COV107-23, COVD21-C8, and CC12.3 contain F58, whereas the WT of COVA2-20 contains Y58. The light chain gene usage and the number of amino acid (aa) residues in the CDR H3 region of each antibody are indicated. Source data are provided in the Source data file.

of the hydroxyl group, the side chain of Y58F moves closer to the backbone carbon of RBD T415 (Supplementary Fig. 11). The average distance between the centroid of the side-chain aromatic ring at $V_H$ residue 58 and the backbone carbon of RBD T415 is 5.3 Å and 5.9 Å for antibodies that carry F58 and Y58, respectively. Since T-shaped π-π stacking is optimal at around 5.0–5.2 Å[60,61], F58 but not Y58 can form strong T-shaped π–π stacking interactions with the amide backbone of RBD T415. This observation can at least partly explain why Y58F improves affinity despite the loss of a hydrogen bond with the RBD.

## Discussion

While several studies to date have described IGHV3-53/3-66 as a commonly used germline for SARS-CoV-2 RBD antibodies[25,33,49,56], the exact sequence requirements for generating an IGHV3-53/3-66 antibody to SARS-CoV-2 RBD has remained largely elusive. As a result of numerous efforts from multiple groups in isolating RBD antibodies and reporting their sequences[22–42], detailed characterization of RBD antibody sequence features has become possible. Through sequence analysis, biophysical experiments, and high-throughput screening, we identified distinct sequence requirements for two public clonotypes (clonotypes 1 and 2) of IGHV3-53/3-66 RBD antibodies. In fact, the frequent occurrence of IGHV3-53/3-66 RBD antibodies with IGHJ6 and a CDR H3 length of 9 amino acids, which are germline features of clonotype 1 antibodies, have also been reported in previous publications[25,62]. Notably, there are several medium-sized clonotypes that are paired with IGKV1-33 and have a CDR H3 of equal to or less than 11 amino acids, which warrant further investigation.

One important finding in this study is that the CDR H3 sequence that supports IGHV3-53/3-66 antibodies binding to RBD is light chain-dependent. This finding is consistent with our previous observation that there is a large diversity of CDR H3 sequences in IGHV3-53/3-66 RBD antibodies[56]. In addition, our findings explain a recent observation by Banach and colleagues[63] who showed that swapping the heavy and light chains of different IGHV3-53/3-66 RBD antibodies often substantially reduced their neutralization potency. Therefore, IGHV3-53/3-66 provides a robust framework to generate different public clonotypes that have distinct CDR H3 and light chain sequence signatures. While only two major clonotypes of IGHV3-53/3-66 RBD antibodies are examined in this study, it will be worth characterizing other minor clonotypes to obtain a more complete understanding of the compatibility between CDR H3 sequence and light-chain identity among IGHV3-53/3-66 RBD antibodies.

As shown in this study (Fig. 1a) and our previous work[49], IGKV1-9 is the most commonly used light chain gene among IGHV3-53/3-66 RBD antibodies. This observation may be attributable to two possible scenarios. Firstly, the affinity of IGHV3-53/3-66 antibodies that use IGKV1-9 may have a higher affinity to RBD on average than those that do not use IGKV1-9. Secondly, IGKV1-9 may be compatible with more diverse CDR H3 sequences than non-IGKV1-9 antibodies (Fig. 1b), which allows IGKV1-9 to be selected more frequently in IGHV3-53/3-66 RBD antibodies. Nevertheless, these speculations require further experimental confirmation.

Although this study revealed that Y58F is a common SHM that improves the affinity of IGHV3-53/3-66 antibodies with a short CDR H3 to RBD, other common SHMs have also shown up in our sequence analysis (Fig. 5a), albeit with a lower frequency. Most noticeably, a cluster of common SHMs is found in $V_H$ framework region 1 from residues 26 to 28. This cluster of SHMs is also likely to be important for affinity maturation to RBD. A recent study has indeed shown that the SHMs F27L and F27V

increase affinity of the antibody to the RBD even though there is a potential loss of π interactions with the antibody[40]. Thus, a relatively less bulky side chain appears to be beneficial at amino acid position 27 for higher affinity binding of the antibody to the RBD. Additional common SHMs among IGHV3-53/3-66 RBD antibodies with a short CDR H3 include S31R in CDR H1 and V50L in CDR H2 (Fig. 5a), which may also play an important role in the affinity maturation of IGHV3-53/3-66 RBD antibodies. As a result, while IGHV3-53/3-66 RBD antibodies do not necessarily require any SHM to neutralize SARS-CoV-2[59], this study along with others have shown that SHM can substantially improve the binding affinity of IGHV3-53/3-66 antibodies to RBD[40,59]. Consistently, RBD antibodies from convalescent SARS-CoV-2 patients have significantly more SHMs and higher neutralization potency at 6-month post-infection than at 1-month post-infection[64].

Circulating SARS-CoV-2 mutant variants represent a major ongoing challenge to natural immunity and vaccination. In particular, a lot of attention has been focused on RBD mutation E484K, which has emerged in multiple independently SARS-CoV-2 lineages[65,66] and can alter the antigenicity of the spike protein[67–69]. Another naturally occurring RBD mutation, K417N/T, which has emerged in South Africa and Brazil (B.1.351 and B.1.1.28 lineages, respectively)[65,66,70], has recently been shown to also alter antigenicity of the spike protein[68,71–73]. We found that K417N dramatically decreased the binding of COV107-23 (clonotype 1) and COVD21-C8 (clonotype 2) to RBD (Supplementary Fig. 12a, b). In fact, K417 forms an electrostatic interaction with the signature residue $V_H$ D/E98 of CDR H3 in clonotype 2 antibodies (Fig. 2b) and can also interact with CDR H3 of clonotype 1 antibodies (Supplementary Fig. 12c), providing a structural explanation for its change in antigenicity. Consistently, IGHV3-53/3-66 RBD antibodies have decreased neutralization activities against the P.1 lineage, which contains a K417T mutation[74]. IGHV3-53/3-66 RBD antibodies also show decreased neutralization activities against the B.1.1.7 lineage[75]. Of note, the B.1.1.7 lineage contains an N501Y mutation in the RBD[75], which is within the epitope of IGHV3-53/3-66 RBD antibodies[76]. For two other lineages of concern, B.1.429 and P.2, the only RBD mutations are L452R and E484K, respectively. Both L452R and E484K are outside the epitope of IGHV3-53/3-66 antibodies. In fact, Li and colleagues demonstrated that the L452R mutation does not weaken the neutralizing activity of IGHV3-53/3-66 RBD antibodies, including B38, CB6, and P2C-1F11[77]. Our recent study also demonstrated that E484K does not alter the neutralizing activity of IGHV3-53/3-66 RBD antibodies with short CDR H3[76]. Constant antigenic drift of SARS-CoV-2 is unavoidable if it keeps circulating among humans. Thus, sustained efforts in characterizing the antibody response to SARS-CoV-2 as it evolves will not only benefit vaccine development and assessment, but also improve our fundamental understanding of the ability of the antibody repertoire to rapidly respond to viral infections.

## Methods

**Literature mining for antibodies to SARS-CoV-2 RBD.** Sequences of anti-SARS-CoV-2 RBD from convalescent patients infected with SARS-CoV-2 were obtained from published articles[22–42] (Supplementary Data 1). IgBLAST was used to identify somatic hypermutations and analyze *IGHJ* gene usage[78]. Of note, IgBLAST can only identify *IGHJ* gene usage for antibodies with available nucleotide sequences. Sequence logos were generated by WebLogo[79].

**Expression and purification of Fc-tagged RBD.** The receptor-binding domain (RBD) (residues 319–541) of the SARS-CoV-2 spike (S) protein (GenBank: QHD43416.1) was fused with an N-terminal Igκ secretion signal and a C-terminal SSSSG linker followed by an Fc tag and cloned into a phCMV3 vector. The plasmid was transiently transfected into Expi293F cells using ExpiFectamine™ 293 Reagent (Thermo Fisher Scientific) according to the manufacturer's instructions. The supernatant was

collected at 7 days post-transfection. The Fc-tagged RBD was purified with by Kan-CapA protein A affinity resin (Kaneka).

**Expression and purification of Fabs**. Fab heavy and light chains were cloned into phCMV3. Heavy chain Y58F or F58Y mutants were constructed using the Quik-Change XL Mutagenesis kit (Stratagene) according to the manufacturer's instructions. The plasmids were transiently co-transfected into Expi293F cells at a ratio of 2:1 (HC:LC) using ExpiFectamine™ 293 Reagent (Thermo Fisher Scientific) according to the manufacturer's instructions. The supernatant was collected at 7 days post-transfection. The Fab was purified with a CaptureSelect™ CH1-XL Prepacked Column (Thermo Fisher Scientific).

**Biolayer interferometry binding assay**. Binding assays were performed by biolayer interferometry (BLI) using an Octet Red96e instrument (FortéBio) at room temperature as described previously[80]. Briefly, Fc-tagged SARS-CoV-2 RBD proteins at 20 µg/ml in 1× kinetics buffer (1× PBS, pH 7.4, 0.01% w/v BSA and 0.002% v/v Tween 20) were loaded onto anti-hIgG Fc Capture (AHC) biosensors and incubated with the indicated concentrations of Fabs. The assay consisted of five steps: (1) baseline: 60 s with 1× kinetics buffer; (2) loading: 300 s with Fc-tagged SARS-CoV-2 RBD proteins; (3) baseline: 60 s with 1× kinetics buffer; (4) association: 60 s with Fab samples; and (5) dissociation: 60 s with 1× kinetics buffer. For estimating the exact $K_D$, a 1:1 binding model was used.

**X-ray crystallography**. Fabs COV107-23 (15 mg/ml) and COV107-23 paired with the light chain of COVD21-C8 (COV107-23-swap, 14 mg/ml) were screened for crystallization using the 384 conditions of the JCSG Core Suite (Qiagen) on our robotic CrystalMation system (Rigaku) at Scripps Research by the vapor diffusion method in sitting drops containing 0.1 µl of protein and 0.1 µl of reservoir solution. For COV107-23, optimized crystals were grown in 0.085 M of sodium citrate - citric acid pH 5.6, 0.17 M ammonium acetate, 15% (v/v) glycerol, and 25.5% (w/v) polyethylene glycol 4000 at 20 °C. For COV107-23-swap, optimized crystals were grown in 0.1 M of sodium citrate pH 4, 1 M lithium chloride, and 20% (w/v) polyethylene glycol 6000 at 20 °C. Crystals were grown for 7 days and then harvested and flash cooled in liquid nitrogen. Diffraction data were collected at cryogenic temperature (100 K) at Stanford Synchrotron Radiation Lightsource (SSRL) on the Scripps/Stanford beamline 12-1 with a beam wavelength of 0.97946 Å, and processed with HKL2000 (version 712)[81]. Structures were solved by molecular replacement using PHASER (version 2.1.2)[82], where the models were generated by Repertoire Builder (https://sysimm.org/rep_builder/)[83]. Iterative model building and refinement were carried out in COOT (version 0.8.9)[84] and PHENIX (version 1.12-2829)[85], respectively.

**Construction of plasmids and CDR H3 library**. Primer names and their sequences are listed in Supplementary Table 2. 143 oligonucleotides (Supplementary Data 3) encoding CDR H3 were obtained from Integrated DNA Technologies (IDT) and PCR-amplified using Oligo-F as forward primer and Oligo-R as reverse primer. Then, the amplified oligonucleotide pool was gel-purified using a GeneJET Gel Extraction Kit (Thermo Scientific).

Wild-type (WT) B38 yeast display plasmid, pCTcon2_B38, was generated by cloning the coding sequence of (from N-terminal to C-terminal, all in-frame) Aga2 secretion signal, B38 Fab light chain, V5 tag, ERBV-1 2A self-cleaving peptide, Aga2 secretion signal, B38 Fab heavy chain, HA tag, and Aga2p, into the pCTcon2 vector[86]. pCTcon2_B38 was PCR-amplified using B38-VF as forward primer and B38-VR as reverse primer to generate the linearized vector. The PCR product was then gel-purified.

**Yeast antibody display library generation**. The B38 yeast antibody display library with different CDR H3 variants was generated following previously published protocol[87]. Saccharomyces cerevisiae EBY100 cells (American Type Culture Collection) were grown in YPD medium (1% w/v yeast nitrogen base, 2% w/v peptone, 2% w/v D(+)-glucose) overnight at 30 °C with shaking at 225 rpm until $OD_{600}$ has reached 3. Then, an aliquot of overnight culture was grown in 100 ml YPD media, with an initial $OD_{600}$ of 0.3, shaking at 225 rpm at 30 °C. Once $OD_{600}$ has reached 1.6, cells were collected by centrifugation at $1700 \times g$ for 3 min at room temperature. Media was removed and the cell pellet was washed twice with 50 ml ice-cold water, and then once with 50 ml of ice-cold electroporation buffer (1 M sorbitol, 1 mM calcium chloride). Cells were resuspended in 20 ml conditioning media (0.1 M lithium acetate, 10 mM dithiothreitol), shaking at 225 rpm at 30 °C. Cells were collected via centrifugation at $1700 \times g$ for 3 min at room temperature, washed once with 50 ml ice-cold electroporation buffer, resuspended in electroporation buffer to reach a final volume of 1 ml, and kept on ice. 5 µg of the amplified oligonucleotide pool and 4 µg of purified linearized vector were added into 400 µl of conditioned yeast. The mixture was transferred to a pre-chilled BioRad GenePulser cuvette with a 2 mm electrode gap and kept on ice for 5 min until electroporation. Cells were electroporated at 2.5 kV and 25 µF, achieving a time constant between 3.7 and 4.1 ms. Electroporated cells were transferred into 4 ml of YPD media supplemented with 4 ml of 1 M sorbitol and incubated at 30 °C

with shaking at 225 rpm for 1 h. Cells were collected via centrifugation at $1700 \times g$ for 3 min at room temperature, resuspended in 0.6 ml SD-CAA medium (2% w/v D-glucose, 0.67% w/v yeast nitrogen base with ammonium sulfate, 0.5% w/v casamino acids, 0.54% w/v $Na_2HPO_4$, 0.86% w/v $NaH_2PO_4 \cdot H_2O$, all dissolved in deionized water), plated onto SD-CAA plates (2% w/v D-glucose, 0.67% w/v yeast nitrogen base with ammonium sulfate, 0.5% w/v casamino acids, 0.54% w/v $Na_2HPO_4$, 0.86% w/v $NaH_2PO_4 \cdot H_2O$, 18.2% w/v sorbitol, 1.5% w/v agar, all dissolved in deionized water) and incubated at 30 °C for 40 h. Colonies were then collected in SD-CAA medium, centrifuged at $1700 \times g$ for 5 min at room temperature, and resuspended in SD-CAA medium with 15% v/v glycerol such that $OD_{600}$ was 50. Glycerol stocks were stored at −80 °C.

**Fluorescence-activated cell sorting of yeast antibody display library**. 100 µl of WT B38 yeast antibody display library glycerol stock was recovered in 50 ml SD-CAA medium by incubating at 27 °C with shaking at 250 rpm until $OD_{600}$ reached between 1.5 and 2.0. At this time, 15 ml of the yeast culture was harvested, and the yeast pellet was obtained via centrifugation at $4000 \times g$ at 4 °C for 5 min. The supernatant was discarded, and SGR-CAA (2% w/v galactose, 2% w/v raffinose, 0.1% w/v D-glucose, 0.67% w/v yeast nitrogen base with ammonium sulfate, 0.5% w/v casamino acids, 0.54% w/v $Na_2HPO_4$, 0.86% w/v $NaH_2PO_4 \cdot H_2O$, all dissolved in deionized water) was added to make up the volume to 50 ml. The yeast culture was then transferred to a baffled flask and incubated at 18 °C with shaking at 250 rpm. Once $OD_{600}$ had reached between 1.3 and 1.6, 1 ml of yeast culture was harvested, and the yeast pellet was obtained via centrifugation at $4000 \times g$ at 4 °C for 5 min. The pellet was subsequently washed with 1 ml of 1× PBS twice. After the final wash, cells were resuspended in 1 ml of 1× PBS.

Then, for expression assay, PE anti-HA.11 (epitope 16B12, BioLegend, Cat. No. 901517) that was buffer-exchanged into 1× PBS was added to the cells at a final concentration of 1 µg/ml. A negative control was set up with nothing added to the PBS-resuspended cells. Samples were incubated overnight at 4 °C with rotation. Then, the yeast pellet was washed twice in 1× PBS and resuspended in FACS tubes containing 2 ml 1× PBS. Using a BD FACS Aria II cell sorter (BD Biosciences) and FACS Diva software v8.0.1 (BD Biosciences), PE-positive cells were collected in 1 ml of SD-CAA containing 1× penicillin/streptomycin. Cells were then collected via centrifugation at $3800 \times g$ at 20 °C for 15 min. The supernatant was discarded. Subsequently, the pellet was resuspended in 100 µl of SD-CAA and plated on SD-CAA plates at 30 °C. After 40 h, colonies were collected in 2 ml of SD-CAA. Frozen stocks were made by reconstituting the pellet in 15% v/v glycerol (in SD-CAA medium) and then stored at −80 °C.

For binding assay, SARS-CoV-2 S RBD-Fc was added to washed cells at a final concentration of 20 µg/ml. A negative control was set up with nothing added to the PBS-resuspended cells. Samples were incubated overnight at 4 °C with rotation. The yeast pellet was then washed twice in 1× PBS. After the last wash, cells were resuspended in 1 ml of 1× PBS. Subsequently, PE anti-human IgG Fc antibody (clone HP6017, BioLegend, Cat. No. 409304) that was buffer-exchanged into 1× PBS was added to yeast at a final concentration of 1 µg/ml. Cells were incubated at 4 °C for 1 h with rotation. The yeast pellet was then washed twice in 1× PBS and resuspended in FACS tubes containing 2 ml 1× PBS. Using a BD FACS Aria II cell sorter (BD Biosciences) and FACS Diva software v8.0.1 (BD Biosciences), PE-positive cells were collected in 1 ml of SD-CAA containing 1× penicillin/streptomycin. Cells were then collected via centrifugation at $3800 \times g$ at 20 °C for 15 min. The supernatant was then discarded. Subsequently, the pellet was resuspended in 100 µl of SD-CAA and plated onto SD-CAA plates at 30 °C. After 40 h, colonies were collected in 2 ml of SD-CAA, and subsequently pelleted. Frozen stocks were made by reconstituting yeast pellets with 15% v/v glycerol (in SD-CAA medium) such that $OD_{600}$ is 50 and then stored at −80 °C.

FCS Express 6 software (De Novo Software) was used to analyze flow cytometry data.

**Next-generation sequencing of CDR H3 loops**. Plasmids from the unsorted yeast display library (input) as well as two replicates of sorted yeast display library based on RBD-binding and expression were extracted from sorted yeast cells using a Zymoprep Yeast Plasmid Miniprep II Kit (Zymo Research) following the manufacturer's protocol. The CDR H3 region was subsequently amplified via PCR using CDRH3-F and CDRH3-R as forward and reverse primers, respectively. Subsequently, adapters containing sequencing barcodes were appended to the genes encoding the CDR H3 region via PCR. 100 ng of each sample was used for paired-end sequencing using Illumina MiSeq PE150 (Illumina). PEAR was used for merging the forward and reverse reads[88]. Regions corresponding to the CDR H3 were extracted from each paired read. The number of reads corresponding to each CDR H3 variant in each sample is counted. A pseudocount of 1 was added to the final count to avoid division by zero in enrichment calculation. The enrichment for variant i was computed using Eq. (1) as follows:

$$\text{Enrichment of variant}_i = \frac{(\text{read count of variant}_i \text{ in sorted sample})/(\text{total read count in sorted sample})}{(\text{read count of variant}_i \text{ in input})/(\text{total read count in input})}$$

(1)

The reported enrichment value for each variant is the average of two biological replicates.

**Reporting summary**. Further information on research design is available in the Nature Research Reporting Summary linked to this article.

## Data availability

Raw sequencing data have been submitted to the NIH Short Read Archive under accession number: BioProject PRJNA691562. Structural data generated in this manuscript have been deposited as 7LK9 and 7LKA to the RCSB Protein Data Bank. Structural data used for analysis are from PDB: 6XC2, 6XC4, 7CH4, 7CH5, 7CJF, 6XC3, 7K8M, 7JMO, 7C01, 6XE1, 7B3O, 7BZ5, 7CDI, 7KFW, 7KFY, 7KFV, and 7KFX. Data for literature mining, biolayer interferometry, and deep sequencing analysis are available at https://github.com/wchnicholas/IGHV3-53_sequence_features, Supplementary Data 1, Supplementary Data 2, and the Source Data files. Biological materials including the wild-type B38 yeast display plasmid, pCTcon2_B38, and the B38 yeast antibody display library are available by contacting the corresponding author (N.C.W.) Source data are provided with this paper.

## Code availability

Custom python scripts for analyzing the deep mutational scanning data have been deposited to https://github.com/wchnicholas/IGHV3-53_sequence_features[89]. Files for simulation using Rosetta Commons (version 3.12) are available at https://github.com/timothyjtan/ighv3-53_3-66_antibody_sequence_features[90].

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

## Acknowledgements
We thank the Roy J. Carver Biotechnology Center at the University of Illinois at Urbana-Champaign for assistance with fluorescence-activated cell sorting and next-generation sequencing. This work was supported by NIH R00 AI139445 (N.C.W.), and Bill and Melinda Gates Foundation INV-004923 (I.A.W.).

## Author contributions
T.J.C.T., G.C.P. and N.C.W. conceived and designed the study. J.R.B., M.Y. and B.M.S. expressed and purified the proteins. T.J.C.T., K.K., X.C., J.R.C. and C.B.B. performed the yeast display experiments. T.J.C.T., Y.W. and N.C.W. processed the next-generation sequencing data. M.Y. and X.Z. performed the crystallization, X-ray data collection, determined and refined the X-ray structures. T.J.C.T., M.Y., K.K., G.C.P., I.A.W. and N.C.W. analyzed the data. T.J.C.T., M.Y., I.A.W. and N.C.W. wrote the paper and all authors reviewed and/or edited the paper.

## Competing interests
The authors declare no competing interests.
