## [Peer Review File · Nature Communications]

REVIEWER COMMENTS

Reviewer #1 (Remarks to the Author):

In this manuscript, the authors collected 214 SARS-CoV-2 RBD binding clonotypes encoded by two very similar IGHV genes (IGHV-53/3-66) by literature mining and analyzed their characteristics. It has been reported that many antibodies encoded by these IGHV genes binds to RBD with no or minimal somatic mutation. But the nature of HCDR3 and light chains not restricting their reactivity to RBD has not been elucidated.

The major findings of this manuscript are as follows.

There are two prominent clonotypes paired with either IGKV1-9 light chain (clonotype 1) or KGKV3-20 (clonotype 2).

These two clonotypes have characteristic HCDR3 of 9 amino acid residues with consensus sequences (Fig. 1a and 1b).

These clonotypes have preference to either IGHJ6 (clonotype 1) or IGHJ4 (clonotype 2) genes (Fig. 1c).

Minor comments:

1) HCDR3 sequences from non-IGKV1-9 antibodies are compatible with a clonotype 1 clone (B38) in constituting mAbs reactive to RBD. It would be great if the authors can speculate the mechanism for the preferential usage of IGKV1-9 over the other genes during the development of B cells encoding clonotype I RBD binding mAbs.

2) Light chain swapping experiment and screening of HCDR3 variants were confined to clonotype 1. But the direct evidence for the interaction between HCDR3 and LCDR3 were found in clonotype 2. If there is a reason(s) for the authors' preference on clonotype 1, please discuss this point.

3) Line 237:

"Our BLI experiments showed that the Y58F mutation dramatically improved the affinity of the three antibodies (COV107-23, COVD21-C8 and CC12.3) by ~10-fold to ~1000-fold (Figure 5b, Supplementary Figure 9)"

Comment) Figure 3a is related to this description. Please add it.

4) It is curious how the codons of seven IGV3-53/66 RBD antibodies with Y58F mutation are composed (a single point mutation at second letter or two consecutive mutations at second and third letters).

Reviewer #2 (Remarks to the Author):

The discovery of public neutralizing antibodies in the sera of patients convalescent for COVID-19 points to 'hardwired' pathways for reproducible elicitation of nAbs against SARS-CoV-2 and are thus of great interest. In the present study, the authors identify additional conserved signatures within the now well-described public nAbs deploying IGHV3-53/3-66-encoded antigen binding loops along with shorter HCDR3s. The authors identify two of the most common B cell clonotypes within a set of 214 published IGHV3-53/3-66 nAbs and describe convergent HCDR3 sequence and structure within each of these groupings. These convergences appear to explain aspects of LC pairing, although the authors also employ yeast display to demonstrate a capacity for swapping the HCDR3s from different clonotypes and maintaining high affinity binding. Finally, the authors demonstrate that many IGHV3-53/3-66 nAbs harbor a public mutation (Y58F) that is reproducibly deployed when the antibodies have a length of less than ~15 amino acids.

General:

This work adds to our understanding of convergent features within human antibody responses against SARS-CoV-2, namely features that are reproducibly deployed to neutralize this virus. The work is timely and significant to the field. The questions listed below should be addressed prior to publication

Questions:

Figure 1. Many clonotypes seem to pair with IGKV1-33 (HCDR3s with <11 amino acids). Why were these not studied?

Figure 4. Do the higher binding variants in B38 also enhance neutralization?

Figure 5a. The authors should modify this figure to include a heatmap column that visually accounts for the total proportions of the 214 IGHV3-53/3-66 nAbs, which if I read this correctly form the dominator in each HCDR3 length category. This would help the reader judge better judge the significance of the public mutation (Y58F and also potentially others, e.g. F27L, F27V, V50L) in the nAbs that are <15 amino acids. I suspect it will strengthen significance, given that nAbs with <15 amino acids are the most abundant, as shown in Figure 1. Visually, Figure 5a gives the strongest heat signals for >15amino acids, but this is likely artificially high due to the more limited number of nAbs with longer lengths.

Figure 5b. Related to above. There appear to be conserved other patterns in SHM for nAbs with <15 amino acids e.g. F27L, F27V, V50L. Authors should comment on the potential significance.

Supplementary Figure 11. K417N is one of the RBD mutations from B.1.351. It will be important to know whether human public IGHV3-53/3-66 nAbs are also comprised by RBD mutations within other strain variants of concern (e.g. B.1.1.7, B.1.1.298, B.1.1.429, P.1, P.2).

Reviewer #3 (Remarks to the Author):

In this study, Tan et al. used literature mining to identify two public clonotypes of IGHV3-53/3-66 antibodies that recognize the RBD of SARS-CoV-2 spike protein. Both clonotypes have a VH CDR3 length of 9 residues, but with distinct sequence motifs. Clonotype 1 was found to pair with IGKV1-9 L chains and clonotype 2 with IGKV3-20 L chains. An analysis of published crystal structures of clonotype 1 and clonotype 2 antibodies in complex with the RBD provided a plausible explanation for the different VH CDR3 sequence motifs of these clonotypes. When the authors swapped the L chains of clonotype 1 and clonotype 2 antibodies, binding to the RBD was reduced >1000-fold. Crystal structures of two L chain-swapped antibodies in apo form showed that VH CDR3 adopted different conformations when paired with different L chains, accounting for the affinity loss. The authors also identified a somatic mutation (Y58F) that increased the affinity of IGHV3-53/3-66 antibodies up to 1000-fold. This study is a useful contribution to on-going efforts to characterize the antibody response to SARS-CoV-2.

Point to address:

To better explain the effect of the Y58F mutation, the authors should perform in silico mutagenesis of the antibody-RBD interface using Rosetta. What is the predicted $\Delta\Delta G$ for the Y58F mutation? Individual Rosetta scoring function terms comprising the predicted $\Delta\Delta G$ might provide mechanistic insights into the effect of this mutation.

Reviewer #1 (Remarks to the Author):

In this manuscript, the authors collected 214 SARS-CoV-2 RBD binding clonotypes encoded by two very similar IGHV genes (IGHV3-53/3-66) by literature mining and analyzed their characteristics. It has been reported that many antibodies encoded by these IGHV genes binds to RBD with no or minimal somatic mutation. But the nature of HCDR3 and light chains not restricting their reactivity to RBD has not been elucidated.

The major findings of this manuscript are as follows.

There are two prominent clonotypes paired with either IGKV1-9 light chain (clonotype 1) or KGKV3-20 (clonotype 2).

These two clonotypes have characteristic HCDR3 of 9 amino acid residues with consensus sequences (Fig. 1a and 1b).

These clonotypes have preference to either IGHJ6 (clonotype 1) or IGHJ4 (clonotype 2) genes (Fig. 1c).

Minor comments:

1) HCDR3 sequences from non-IGKV1-9 antibodies are compatible with a clonotype 1 clone (B38) in constituting mAbs reactive to RBD. It would be great if the authors can speculate the mechanism for the preferential usage of IGKV1-9 over the other genes during the development of B cells encoding clonotype I RBD binding mAbs.

Response: Thank you for the comment. The mechanism for the preferential usage of IGKV1-9 over the other light chain genes in IGHV3-53/3-66 RBD antibodies is speculated in the discussion section of the revised manuscript:

“As shown in this study (Supplementary Table 1) and our previous work⁴⁹, IGKV1-9 is the most commonly used light chain gene among IGHV3-53/3-66 RBD antibodies. This observation may be attributable to two possible scenarios. Firstly, the affinity of IGHV3-53/3-66 antibodies that use IGKV1-9 may have a higher affinity to RBD on average than those that do not use IGKV1-9. Secondly, IGKV1-9 may be compatible with more diverse CDR H3 sequences than non-IGKV1-9 antibodies (Figure 1b), which allows IGKV1-9 to be selected more frequently in IGHV3-53/3-66 RBD antibodies. Nevertheless, these speculations require further experimental confirmation.”

2) Light chain swapping experiment and screening of HCDR3 variants were confined to clonotype 1. But the direct evidence for the interaction between HCDR3 and LCDR3 were found in clonotype 2. If there is a reason(s) for the authors' preference on clonotype 1, please discuss this point.

Response: In fact, the swapping experiment was performed for both clonotypes 1 and 2 (Figure 3a). We have clarified this in the figure legend by adding the following:

“COV107-23 belongs to clonotype 1, whereas COVD21-C8 belongs to clonotype 2.”

Additionally, we selected clonotype 1 for CDR H3 screening because it is the predominant clonotype. We have now added the following in the results section of the revised manuscript:

“In particular, we focused on identifying CDR H3 sequences that are compatible with IGKV1-9, which is used by clonotype 1 antibodies for binding to RBD, because IGKV1-9

is the most commonly used light chain gene among IGHV3-53/3-66 RBD antibodies (Supplementary Table 1) and clonotype 1 is the most predominant clonotype (Figure 1a).”

3) Line 237:

“Our BLI experiments showed that the Y58F mutation dramatically improved the affinity of the three antibodies (COV107-23, COVD21-C8 and CC12.3) by ~10-fold to ~1000-fold (Figure 5b, Supplementary Figure 9)”

Comment) Figure 3a is related to this description. Please add it.

Response: We have now added “Figure 3a” as highlighted in the text.

4) It is curious how the codons of seven IGV3-53/66 RBD antibodies with Y58F mutation are composed (a single point mutation at second letter or two consecutive mutations at second and third letters).

Response: Using IgBLAST, we identified that the codons of three of the seven IGHV3-53/3-66 RBD antibodies (CC12.1, CC12.3, CB6) with the Y58F mutation have a single point mutation from a thymine to an adenine at the second nucleotide. The germline sequence of IGHV3-53/3-66 has the codon TAC that encodes Y58, while CC12.1, CC12.3 and CB6 have the codon TTC that encodes F58. The other four antibodies – P4A1, C102, BD-604 and COVA2-04 – do not have their sequences published. We have added the following to the figure legend of Supplementary Figure 11:

“Nucleotide sequences are publicly available for three antibodies with Y58F, namely CC12.1, CC12.3 and CB6^{4,5}. For these three antibodies, Y58F occurs through a single point mutation in the second nucleotide of the codon as shown by IgBLAST analysis⁶. Specifically, the germline IGHV3-53/3-66 gene contains TAC, which encodes Y58, while the antibodies CC12.1, CC12.3 and CB6 contain TTC, which encodes F58.”

Reviewer #2 (Remarks to the Author):

The discovery of public neutralizing antibodies in the sera of patients convalescent for COVID-19 points to 'hardwired' pathways for reproducible elicitation of nAbs against SARS-CoV-2 and are thus of great interest. In the present study, the authors identify additional conserved signatures within the now well-described public nAbs deploying IGHV3-53/3-66-encoded antigen binding loops along with shorter HCDR3s. The authors identify two of the most common B cell clonotypes within a set of 214 published IGHV3-53/3-66 nAbs and describe convergent HCDR3 sequence and structure within each of these groupings. These convergences appear to explain aspects of LC pairing, although the authors also employ yeast display to demonstrate a capacity for swapping the HCDR3s from different clonotypes and maintaining high affinity binding. Finally, the authors demonstrate that many IGHV3-53/3-66 nAbs harbor a public mutation (Y58F) that is reproducibly deployed when the antibodies have a length of less than ~15 amino acids.

General:

This work adds to our understanding of convergent features within human antibody responses against SARS-CoV-2, namely features that are reproducibly deployed to neutralize this virus. The work is timely and significant to the field. The questions listed below should be addressed prior to publication

Response: We thank the reviewer for the positive and helpful comments.

Questions:

Figure 1. Many clonotypes seem to pair with IGKV1-33 (HCDR3s with <11 amino acids). Why were these not studied?

Response: Since we only selected the two most frequent clonotypes, which were defined as having the same IGKV gene usage and the same CDR H3 length, we have not included clonotypes that pair with IGKV1-33 in our study. However, the observation mentioned by the reviewer is definitely interesting, which is described in the discussion section of the revised manuscript:

“Notably, there are several medium-sized clonotypes that are paired with IGKV1-33 and have a CDR H3 of equal to or less than 11 amino acids, which warrant further investigation.”

Figure 4. Do the higher binding variants in B38 also enhance neutralization?

Response: Thank you for the comment. We agree that the correlation between binding and neutralization potency is an interesting point to consider. In the revised manuscript, this issue is mentioned in the results section along with a new Supplementary Figure 9:

“Of note, our previous work has shown that binding affinity correlates well with neutralization activity for antibodies that bind to the epitope of B38 (i.e. epitope RBD-A, see Figure 4G in Rogers et al.²⁶). In addition, the binding affinity and neutralization activity of five clonotype 1 antibodies from Cao et al.²⁵ show a high correlation (R = 0.86, Supplementary Figure 9). As a result, although the neutralization potency of B38 variants was not measured in this study, B38 variants with higher binding affinity would likely result in higher neutralization potency.”

Figure 5a. The authors should modify this figure to include a heatmap column that visually accounts for the total proportions of the 214 IGHV3-53/3-66 nAbs, which if I read this correctly form the denominator in each HCDR3 length category. This would help the reader judge better the significance of the public mutation (Y58F and also potentially others, e.g. F27L, F27V, V50L) in the nAbs that are <15 amino acids. I suspect it will strengthen significance, given that nAbs with <15 amino acids are the most abundant, as shown in Figure 1. Visually, Figure 5a gives the strongest heat signals for >15 amino acids, but this is likely artificially high due to the more limited number of nAbs with longer lengths.

Response: Thank you for the suggestion. The reviewer is correct that the strong signals for >15 amino acids are due to the fewer number of nAbs with longer CDR H3. In the revised Figure 5a, the number of nAbs in each category is indicated. We have also added a sentence in the figure legend accordingly:

“The number of antibodies in each category is indicated on the right of the heatmap.”

Also, our analysis was only performed on 165 nAbs that have sequences available instead of 214 nAbs. We mistakenly indicated that the analysis was performed on 214 nAbs, which is corrected in the results section of the revised manuscript:

“This analysis included 165 IGHV3-53/3-66 RBD antibodies that have sequence information available.”

Figure 5b. Related to above. There appear to be conserved other patterns in SHM for nAbs with <15 amino acids e.g. F27L, F27V, V50L. Authors should comment on the potential significance.

Response: In the revised manuscript, we have further discussed the importance of other SHMs in the discussion section:

“Most noticeably, a cluster of common SHMs is found in V_H framework region 1 from residues 26 to 28. This cluster of SHMs is also likely to be important for affinity maturation to RBD. A recent study has indeed shown that the SHMs F27L and F27V increase affinity of the antibody to the RBD even though there is a potential loss of π interactions with the antibody⁴⁰. Thus, a relatively less bulky side chain appears to be beneficial at amino acid position 27 for higher affinity binding of the antibody to the RBD. Additional common SHMs among IGHV3-53/3-66 RBD antibodies with a short CDR H3 include S31R in CDR H1 and V50L in CDR H2 (Figure 5a), which may also play an important role in the affinity maturation of IGHV3-53/3-66 RBD antibodies.”

Supplementary Figure 11. K417N is one of the RBD mutations from B.1.351. It will be important to know whether human public IGHV3-53/3-66 nAbs are also comprised by RBD mutations within other strain variants of concern (e.g. B.1.1.7, B.1.1.298, B.1.1.429, P.1, P.2).

Response: We agree with the reviewer that it is important to also consider those variants of concern. We have now added the following to the discussion section:

“Consistently, IGHV3-53/3-66 RBD antibodies have decreased neutralization activities against the P.1 lineage, which contains a K417T mutation⁷⁴. IGHV3-53/3-66 RBD antibodies also show decreased neutralization activities against the B.1.1.7 lineage⁷⁵. Of note, the B.1.1.7 lineage contains an N501Y mutation in the RBD⁷⁵, which is within the epitope of IGHV3-53/3-66 RBD antibodies⁷⁶. For the two other lineages of concern,

B.1.429 and P.2, the only RBD mutations are L452R and E484K, respectively. Both L452R and E484K are outside the epitope of IGHV3-53/3-66 antibodies. In fact, Li and colleagues demonstrated that the L452R mutation does not weaken the neutralizing activity of IGHV3-53/3-66 RBD antibodies, including B38, CB6, and P2C-1F11⁷⁷. Our recent study also demonstrated that E484K does not alter the neutralizing activity of IGHV3-53/3-66 RBD antibodies⁷⁶.”

We recently also obtained data showing that the B.1.1.298 lineage, which has a Y453F mutation in the RBD, does not affect the binding of IGHV3-53/3-66 antibodies. However, we prefer to publish these data in a separate manuscript.

Reviewer #3 (Remarks to the Author):

In this study, Tan et al. used literature mining to identify two public clonotypes of IGHV3-53/3-66 antibodies that recognize the RBD of SARS-CoV-2 spike protein. Both clonotypes have a VH CDR3 length of 9 residues, but with distinct sequence motifs. Clonotype 1 was found to pair with IGKV1-9 L chains and clonotype 2 with IGKV3-20 L chains. An analysis of published crystal structures of clonotype 1 and clonotype 2 antibodies in complex with the RBD provided a plausible explanation for the different VH CDR3 sequence motifs of these clonotypes. When the authors swapped the L chains of clonotype 1 and clonotype 2 antibodies, binding to the RBD was reduced >1000-fold. Crystal structures of two L chain-swapped antibodies in apo form showed that VH CDR3 adopted different conformations when paired with different L chains, accounting for the affinity loss. The authors also identified a somatic mutation (Y58F) that increased the affinity of IGHV3-53/3-66 antibodies up to 1000-fold. This study is a useful contribution to on-going efforts to characterize the antibody response to SARS-CoV-2.

Response: We thank the reviewer for the encouraging comments.

Point to address:

To better explain the effect of the Y58F mutation, the authors should perform in silico mutagenesis of the antibody-RBD interface using Rosetta. What is the predicted $\Delta\Delta G$ for the Y58F mutation? Individual Rosetta scoring function terms comprising the predicted $\Delta\Delta G$ might provide mechanistic insights into the effect of this mutation.

Response: We performed Rosetta simulation using the CC12.3 antibody, which has F58 in the wild type. We generated an F58Y mutant (i.e. germline revertant) of CC12.3 using the 'fixed backbone' protocol, and performed fast relax on both the wild type and mutant antibodies. To incorporate π interactions, we used scoring weights as described in Combs et al. (PMID: 29641200). By comparing the results of wild type CC12.3 (average score = -1987.244 ; $n = 30$) and its germline revertant (average score = -1977.274 ; $n = 30$), the predicted $\Delta\Delta G$ for the somatic hypermutation Y58F is -9.97 (in Rosetta energy units), suggesting that somatic hypermutation Y58F enhances the stability of the antibody-RBD complex. However, the result is not statistically significant ($p = 0.19$, one-tailed t-test). Additionally, the π - π interaction scores for F58 and Y58 are -7.035 and -6.868 , respectively, indicating that F58 increases the π - π interaction as suggested by our structural analysis (Supplementary Figure 11). However, this result is also not statistically significant ($p = 0.10$, one-tailed t-test). Given that Rosetta simulation does not fully recapitulate our experimental binding data, which shows that somatic hypermutation Y58F significantly enhances the binding affinity, we prefer not to include the results of Rosetta simulation in the manuscript. In fact, a benchmark study has shown that Rosetta simulation for $\Delta\Delta G$ prediction only have moderate correlation ($R = 0.5$ to 0.7) with experimental data (Kellogg et al., PMID: 21287615).

REVIEWERS' COMMENTS

Reviewer #1 (Remarks to the Author):

The authors accurately and completely responded to the recommendations. Now I think that the manuscript is acceptable for publication.

Reviewer #2 (Remarks to the Author):

The authors have adequately addressed the questions raised by this reviewer.

Reviewer #3 (Remarks to the Author):

The authors have responded satisfactorily to the previous critiques.

REVIEWERS' COMMENTS

Reviewer #1 (Remarks to the Author):

The authors accurately and completely responded to the recommendations. Now I think that the manuscript is acceptable for publication.

Response: We thank the reviewer for their positive comment.

Reviewer #2 (Remarks to the Author):

The authors have adequately addressed the questions raised by this reviewer.

Response: We thank the reviewer for their positive comment.

Reviewer #3 (Remarks to the Author):

The authors have responded satisfactorily to the previous critiques.

Response: We thank the reviewer for their positive comment.